# Validation of the brief physical activity assessment tool: Comparison of telephone and in-person administration

Rui Vilarinho[1,2☯], Liliana Amorim[3☯], Diana Gomes[4], Pedro Teixeira[3,5], Ana Alves da Silva[4], Janete Santos[4], Filipa Bernardo[6], Jaime Correia de Sousa[5], João Almeida Fonseca[7], Cristina Jácome[7]*

1 FP-I3ID, Escola Superior de Saúde Fernando Pessoa, Porto, Portugal, 2 Centro de Investigação em Reabilitação (CIR), Escola Superior de Saúde, Instituto Politécnico do Porto, Porto, Portugal, 3 Association P5 Digital Medical Center (ACMP5), Braga, Portugal, 4 Center for Health Technology and Services Research (CINTESIS), Faculty of Medicine, University of Porto, Porto, Portugal, 5 Life and Health Sciences Research Institute (ICVS), School of Medicine, ICVS/3Bs, PT Government Associate Laboratory, University of Minho, Braga, Portugal, 6 AstraZeneca, Lisboa, Portugal, 7 CINTESIS@RISE, MEDCIDS, Faculty of Medicine of the University of Porto, Porto, Portugal

☯ These authors contributed equally to this work.
* cjacome@med.up.pt

**Data Availability Statement:** All data supporting the findings are available from the Faculty of Medicine of University of Porto for all interested researchers who meet the criteria for access to

## Abstract

We examined the reliability and validity of the Brief Physical Activity Assessment Tool (BPAAT) when administered by telephone interview compared to in-person administration. We analyzed data from the Epi-asthma study. Adult participants registered in the participating Portuguese primary health care centres (PCC) completed the BPAAT via telephone. After ~3 days (range 0–5 days), they had a face-to-face visit at their PCC and completed BPAAT using a tablet. The BPAAT classify individuals as "insufficiently active" (score 0–3) or "sufficiently active" (score 4–8). 355 subjects (60.8% female, 54[IQR 42–66] years) were included. The median BPAAT score was 2[0–4] for both methods, with a significant correlation (rho = 0.58, p<0.001). Test-retest reliability was moderate (ICC = 0.56, 95%CI 0.49–0.63). Agreement in physical activity classification was fair (71.5%, kappa = 0.31, 95%CI 0.21–0.41), with telephone administration classifying more individuals as "sufficiently active" (37.2%) than in-person (15.5%). Telephone administration of the BPAAT is a valid and reliable approach for monitoring of physical activity in the general population. However, it may slightly overestimate activity levels compared to face-to-face administration, particularly among subjects aged 65 years and older.

## Introduction

Regular physical activity is a well-established protective factor in the prevention and management of the leading noncommunicable diseases, such as cardiovascular diseases, respiratory diseases and cancer [1]. However, nearly a third of adults worldwide (28%) do not meet the World Health Organization's (WHO) global recommendations for physical activity for health

confidential data. Because these data include patients' personal information, the Ethical Committees that approved the study does not recommend that such data be made public unnecessarily. Please contact Cristina Jácome, the corresponding author at cjacome@med.up.pt, or the Faculty of Medicine of University of Porto at fmup@med.up.pt to request the data.

**Funding:** This study was sponsored and funded by AstraZeneca. The funders had no role in study design, data collection and analysis, decision to publish, or preparation of the manuscript.

**Competing interests:** The Authors declare that there is no conflict of interest.

[2]. To address this, a target has been set to reduce physical inactivity by 15% by 2030, including a recommendation to integrate physical activity assessment and counselling into primary health care services [3].

Self-reported questionnaires are attractive for the assessment of physical activity because they are a feasible and simple approach to rapidly screen patients in real-world clinical practice, while being valid when compared to the objective measures [4, 5]. In addition, they can be administered remotely [6] as telephone screening is a key element of hybrid healthcare models [7]. Telephone interviewing has been shown to be comparable to face-to-face administration of self-report questionnaires in general older population and specific clinical populations, such as patients with musculoskeletal disorders and chronic obstructive pulmonary disease [8–10]. Various physical activity screening questionnaires exist, but the Brief Physical Activity Assessment Tool (BPAAT) has been specifically developed to enable healthcare providers to identify inactive patients in primary care settings. It is a simple, quick (<5min) and valid questionnaire in patients with various health conditions [11, 12]. However, its telephone administration has not been studied. Furthermore, existing research on physical activity questionnaires has primarily focused on young and middle-aged adults, leaving a gap in understanding the psychometric properties of these tools in older adults [13, 14]. Therefore, we aimed to determine the reliability and validity of the BPAAT administered by telephone interview in adult subjects from the general population. As secondary aim, we compared the BPAAT performance across young, middle aged and older adults.

## Methods

Data collected between May 2021 and July 2023 from a Portuguese population-based study was analysed [15]. EPI-ASTHMA was investigating the prevalence of current asthma in the adult population [15]. Access to data was provided in November 30th, 2023. Subjects from the general population were included if they were ≥18 years of age and were registered in the database of the participating primary healthcare centres located throughout mainland Portugal. Exclusion criteria were individuals with any specific physical and/or cognitive disabilities that prevented them from cooperating with the study procedures (e.g., answering the questionnaires). The study was approved by the Ethic Committees of the participating centres and participants gave both verbal and written consent to participate. Subjects completed the Portuguese version of BPAAT [4] during a telephone interview and, after approximately 3 days (minimum half day, maximum 5 days), during a face-to-face visit in his/her primary care centre. Interviews were performed by a team of experienced interviewers (psychologists, nurses) using a Computer Assisted Telephone Interviews system (CATI). During the face-to-face visit, participants completed the BPAAT on a tablet, with support available from a trained clinical researcher upon request. The BPAAT consists of two questions, one on the frequency and duration of vigorous-intensity physical activity and the other on moderate-intensity physical activity and walking during a person's usual week. The total score varies from 0 to 8 and allows the individual to be classified as "insufficiently active" (score ≤3) or "sufficiently active" (score ≥ 4) [4, 11]. Convergent validity (Spearman correlation coefficient, rho), reliability (intraclass correlation coefficient [ICC], Bland-Altman analysis), and agreement (% agreement, Cohen κ) were determined for the all sample and across defined age groups (young adults 18–44 years, middle aged adults 45–64 years, older adults ≥65 years) [13, 14]. Spearman correlation coefficient ranges from −1 to 1, with values approaching 0 showing the least correlation [16]. The ICC ranges from 0 to 1 and 'good' to 'excellent' reliability can be considered for ICC>0.75 [17]. The agreement level was interpreted as follows: slight (0.0–0.20), fair (0.21–0.40), moderate (0.41–0.60), substantial (0.61–0.80) or almost perfect (≥0.81) [17].

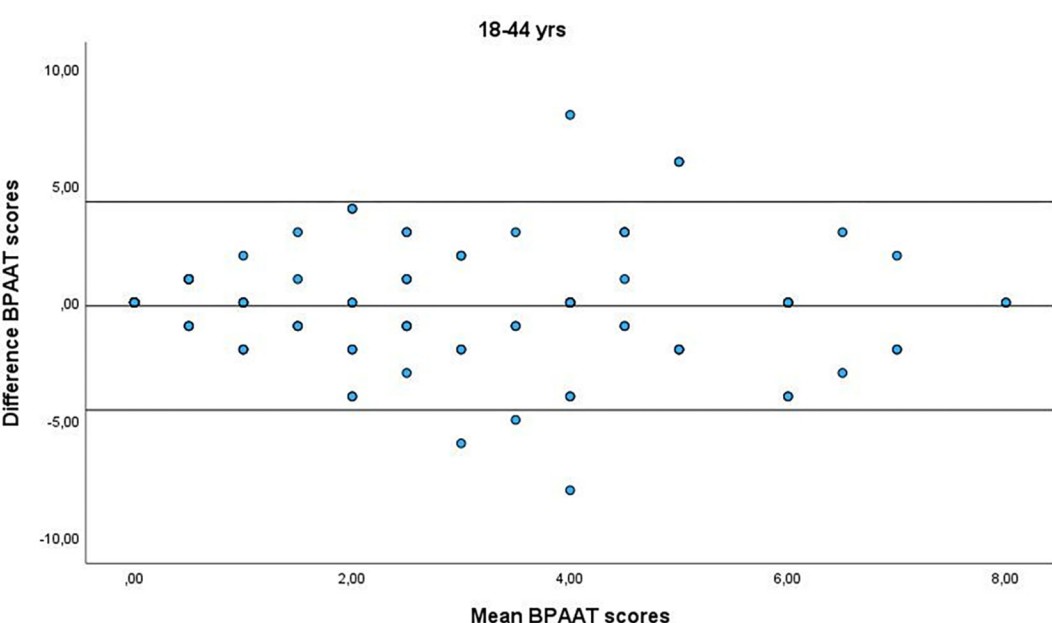

**Fig 1. Bland-Altman plots of the Brief physical activity assessment tool (BPAAT) for subjects aged 18–44 years.** The solid lines represent the mean difference and the 95% limits of agreement.

## Results

A total of 355 subjects (60.8% female) with a median of 54 [interquartile range IQR 42–66] years were analysed. The two BPAATs were completed on the same day in 17.5% of patients (median, 3 [IQR 1–4] days). The median total score was 2 [IQR 0–4] for both methods, which correlated significantly (rho = 0.58; p<0.001). The relative test-retest reliability of the BPAAT scores was moderate (ICC = 0.56, 95%CI 0.49–0.63). There was fair agreement between the versions, with bias close to zero and reasonable limits of agreement (Figs 1–4).

In terms of physical activity classification, the telephone BPAAT classified 223 (62.8%) patients as "insufficiently active" and 132 (37.2%) as "sufficiently active", whereas the tablet version classified 300 (84.5%) as "insufficiently active" and 55 (15.5%) as "sufficiently active". The agreement for physical activity classification was fair (71.5%; Kappa = 0.31, 95%CI 0.21–0.41) (Table 1).

When comparing age groups, the median total score for both methods was 2 [IQR 0–4] in those aged 18–44 years and ≥65 years, compared with a median score of 1 [IQR 0, 4] for those aged 45–64 years. The total scores obtained by the two methods were significantly correlated in the 3 age groups (rho = 0.53–0.64), although with slightly better relative and absolute reliability for the younger age groups (Table 1). When analysing the three age groups, the same pattern was observed in all: telephone administration produced a higher frequency of "sufficiently active" subjects compared to face-to-face administration, although this pattern was more evident in the group aged ≥65 years. This age group also showed the lowest agreement in the physical activity classification (% agreement 61.4%, Kappa = 0.15, 95%CI 0.03–0.29).

## Discussion

The findings from this study suggest that the BPAAT administered by telephone interview demonstrates moderate reliability and validity compared to face-to-face administration in adults from a general population. Furthermore, the fair agreement in physical activity

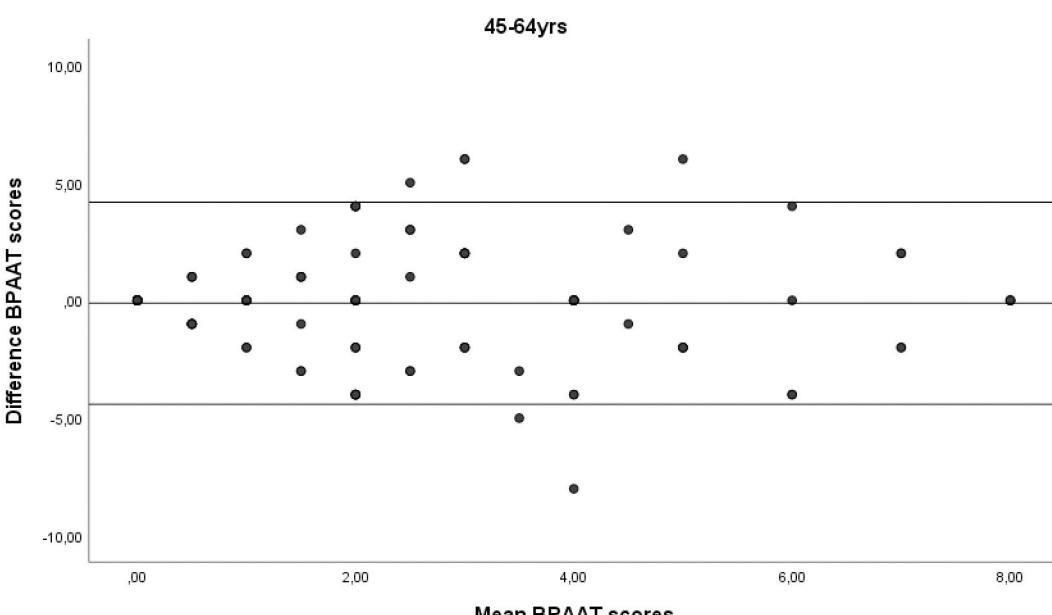

**Fig 2. Bland-Altman plots of the Brief physical activity assessment tool (BPAAT) for subjects aged 45–64 years.** The solid lines represent the mean difference and the 95% limits of agreement.

classification suggests that the two modes of administration are not completely interchangeable. This is in contrast to other established results from other physical activity questionnaires (e.g., the International Physical Activity Questionnaire [IPAQ] [18]) and other self-reported health questionnaires [8–10] in other populations, which report higher reliability and agreement. The higher number of questionnaire items, consistent time intervals, and/or randomisation between the modes of administration may be potential reasons for these differences.

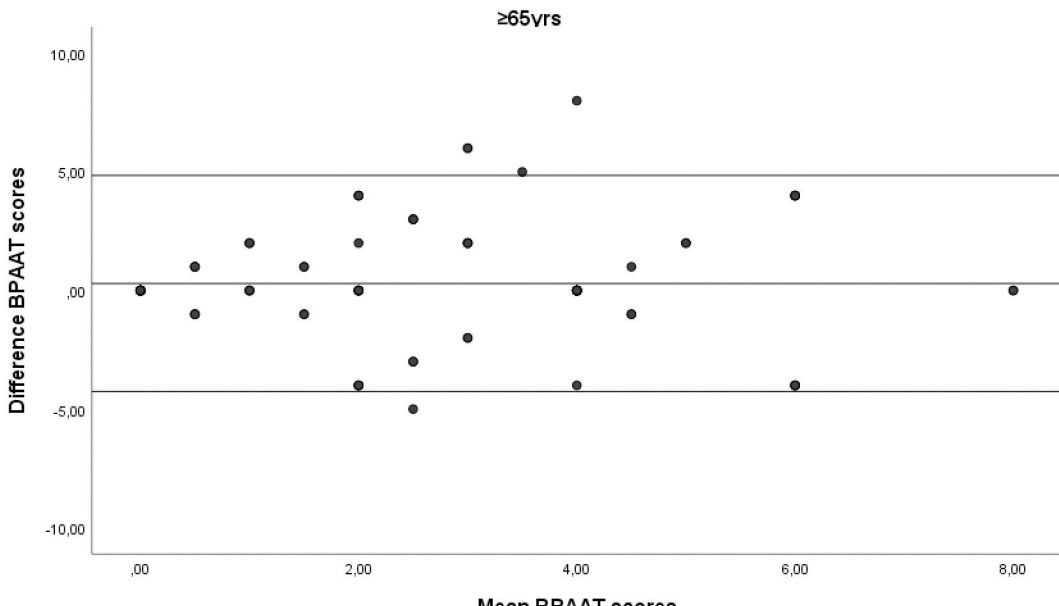

**Fig 3. Bland-Altman plots of the Brief physical activity assessment tool (BPAAT) for subjects aged ≥65 years.** The solid lines represent the mean difference and the 95% limits of agreement.

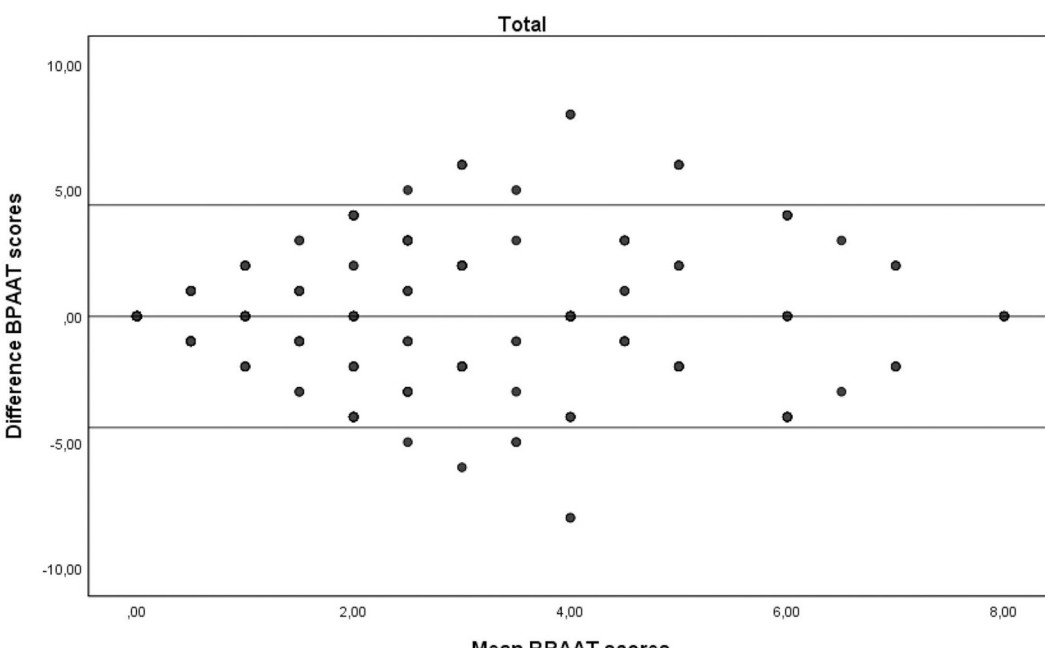

**Fig 4. Bland-Altman plots of the Brief physical activity assessment tool (BPAAT) for all subjects.** The solid lines represent the mean difference and the 95% limits of agreement.

The study highlights the feasibility of using telephone-administered BPAAT in routine clinical practice, particularly given the increasing reliance on remote health services. This method offers a practical approach to screening for physical inactivity, which is consistent with the World Health Organization's recommendations to integrate physical activity assessment into primary care to reduce physical inactivity worldwide. However, the discrepancy in the classification of physical activity levels between the two modes, with higher percentages of "sufficiently active" classifications by telephone, calls for caution. This difference was particularly pronounced in the oldest age group (≥65 years), where the agreement on physical activity classification was the lowest (61.4%, kappa = 0.15). This may be due to differences in how respondents perceive [19] and report their activity levels between the two modes of administration. Telephone interviews may elicit more socially desirable responses [20], leading to a possible

**Table 1. Brief physical activity assessment tool (BPAAT) scores and classification, convergent validity, relative test-retest reliability and agreement (percentage and Cohen k).**

| | BPAAT score telephone | BPAAT score face-to-face | rho | ICC (95% CI) | BPAAT classification "sufficiently active" telephone | BPAAT classification "sufficiently active" face-to-face | % agreement | Cohen k |
|---|---|---|---|---|---|---|---|---|
| **18–44 years** (n = 111) | 2 [0–4] | 2 [0–4] | 0.64* | 0.60 (0.46, 0.70) | 36.9% | 21.6% | 75.7% | 0.43 (0.27, 0.59) |
| **45–64 years** (n = 143) | 1 [0–4] | 1 [0–4] | 0.55* | 0.57 (0.45, 0.67) | 32.2% | 14.7% | 75.5% | 0.35 (0.17, 0.50) |
| **≥65 years** (n = 101) | 2 [0–4] | 2 [0–4] | 0.53* | 0.51 (0.35, 0.65) | 44.6% | 9.9% | 61.4% | 0.15 (0.03, 0.29) |
| **Total** (n = 355) | 2 [0–4] | 2 [0–4] | 0.58* | 0.56 (0.49, 0.63) | 37.2% | 15.5% | 71.5% | 0.31 (0.21, 0.41) |

rho, Spearman correlation coefficient; ICC, intraclass correlation coefficient; CI, confidence interval; *p<0.001

overestimation of physical activity levels. On the other hand, the use of a tablet during the face-to-face visit allowed for assistance/support in reading the BPAAT if requested by the subject, which may have introduced some bias. This suggests that healthcare providers should consider additional methods or follow-up assessments to ensure accurate physical activity classification. This is crucial for the effective tailoring of interventions, especially for populations that may underreport physical inactivity during remote assessments.

There are further limitations to this study. As mentioned above, the lack of random order of the administration modes may have influenced our results, as it did not guarantee that participants had the same opportunity to answer both formats in the first assessment. The irregular time interval between the two modes of administration (range 0–5 days) could also explain the reliability results we found. For example, 17.5% of participants completed both formats on the same day. In future studies, a consistent time interval between formats is important to facilitate accurate interpretations. The inclusion of the conventional (paper) administration of the BPPAT should also be considered. This will provide insight into whether data collected by different methods accurately represent the same constructs, allowing researchers to draw meaningful conclusions and make informed decisions based on the findings. In addition, equivalence allows for greater flexibility in data collection, participant preferences and circumstances, while maintaining the rigour of data collection.

In conclusion, telephone administration of the BPAAT is a valid and reliable approach for monitoring of physical activity in the general population. However, it may slightly overestimate activity levels compared to face-to-face administration, particularly among subjects aged 65 years and older. Despite this, telephone-based assessments represent a valuable tool for remote monitoring of physical activity, offering a convenient and effective supplement to traditional methods.

## Author Contributions

**Conceptualization:** Liliana Amorim, Pedro Teixeira, Janete Santos, Filipa Bernardo, Jaime Correia de Sousa, João Almeida Fonseca, Cristina Jácome.

**Formal analysis:** Rui Vilarinho, Liliana Amorim, Ana Alves da Silva, Janete Santos, Cristina Jácome.

**Funding acquisition:** Jaime Correia de Sousa, João Almeida Fonseca.

**Methodology:** Liliana Amorim, Diana Gomes, Pedro Teixeira, Ana Alves da Silva, Janete Santos, Filipa Bernardo.

**Project administration:** Filipa Bernardo, Jaime Correia de Sousa, João Almeida Fonseca, Cristina Jácome.

**Writing – original draft:** Rui Vilarinho, Liliana Amorim.

**Writing – review & editing:** Rui Vilarinho, Liliana Amorim, Diana Gomes, Pedro Teixeira, Ana Alves da Silva, Janete Santos, Filipa Bernardo, Jaime Correia de Sousa, João Almeida Fonseca, Cristina Jácome.

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
