## [Decision Letter · Decision Letter 0]

12 Nov 2024

PONE-D-24-41112Validation of the Brief Physical Activity Assessment Tool: Comparison of Telephone and In-Person AdministrationPLOS ONE

Dear Dr. Jácome,

Thank you for submitting your manuscript to PLOS ONE. After careful consideration, we feel that it has merit but does not fully meet PLOS ONE’s publication criteria as it currently stands. Therefore, we invite you to submit a revised version of the manuscript that addresses the points raised during the review process.

We look forward to receiving your revised manuscript.

Kind regards,

Filipe Prazeres, MD, MSc, Ph.D.

Academic Editor

PLOS ONE

“This study was sponsored and funded by AstraZeneca.”

Reviewers' comments:

Reviewer's Responses to Questions

**Comments to the Author**

1. Is the manuscript technically sound, and do the data support the conclusions?

Reviewer #1: Yes

Reviewer #2: Yes

Reviewer #3: Partly

2. Has the statistical analysis been performed appropriately and rigorously? 

Reviewer #1: Yes

Reviewer #2: Yes

Reviewer #3: Yes

3. Have the authors made all data underlying the findings in their manuscript fully available?

Reviewer #1: Yes

Reviewer #2: Yes

Reviewer #3: No

4. Is the manuscript presented in an intelligible fashion and written in standard English?

Reviewer #1: Yes

Reviewer #2: Yes

Reviewer #3: Yes

5. Review Comments to the Author

Reviewer #1: General comments

Thank you for giving me the opportunity to review this manuscript. The scope of the study is relevant to the readers of PlosOne and the research field has relevance to assessment of physical activity. The study has a robust design, and the manuscript is generally well written, although blank spaces are missing here and there.

Title

Adequate

Abstract

Adequate

Introduction and aim

The rationale for dividing the sample into age-groups needs to be addressed in the introduction and included in the aim.

Please provide a reference for the first sentence in the introduction.

Materials and methods

State why the age-groups were chosen. State the diary number for the ethical approval.

Results

Why is BMI for the group shown? Make sure only results that is included in the aim is shown, for example the group comparison.

Page 4, line 78: state that figure 1 is figure 1a-d.

Discussion

Well written

Conclusion

Adequate

Tables and figures

Adequate

References

Up-to date and relevant.

Reviewer #2: First of all, I would like to congratulate the authors for this interesting study. It is well-written, clear and addresses an important topic in the research field. I have however some amendments to improve the manuscript:

• Introduction, lines 38-39: “Regular… diseases.” Please add a reference that supports this quote.

• Introduction, lines 48-49: “several clinical populations”. Please provide some examples of these clinical populations. The same for lines 52-53 “in patients with various health conditions”.

• Methods: were the population composed by health individuals, individuals with any medical condition or both? Please clarify this.

• Methods: Please clarify to the readers that the face-to-face version of BPAAT was applied via tablet.

• Methods: Regarding the individuals who performed the BPAAT via phone and in-person on the same day, how did the authors manage it? How long before the face-to-face visit was the phone call made? Could you clarify it? Maybe consider adding a figure…

• Methods: lines 69-71 – please include the interpretation used for the scores of rho, ICC and Cohen K.

• Results: in figure 1, the authors divided the analysis by age decades. Could you explain the reason and clarify this division in the methods?

• Results: figure 1 and table 1 – In the paper from Cruz et al (2021), the results varied from gender. Even though the measures from Cruz et al and this paper are different (accelerometry vs BPAAT in Cruz and only BPAAT administered by different ways in this paper), did the authors consider analyzing the data by gender?

Reviewer #3: It is an interesting study and could be useful practically for researchers or professionals using this scale, and some other scales. It is also interesting to see the differences according to age.

My specific comments are below:

Some of the references are written with a space after the text (like in line.46: "measures (Cleland") but some without space (like in l.47: "remotely(Pleguezuelos") thus they should be standardized, conforming the journal's requirements.

Line 47: "Telephone screening" could be described more openly, since it can also mean filling out a form by clicking its link/QR code etc on a smartphone...

Line 56: Aim: Which general population, it could be specified like all of Portugal or some parts of it, etc.

Methods:

So the questionnaire was applied face-to-face, 3 days after the telephone interview. So the data were collected in this manner between May 2021-July 2023? Or were they contacted later on, like after November 2023, as the original study was on asthma?

Was the ethical approval for the asthma study? Or the physical activity study? Could the ethical committee's decision numbers be provided? (I saw later detailed info in the attachment, ı wonder if just the names and number be stated somewhere but as it is on asthma, maybe the name of the study, the aim and localities with the no.s in parentheses could be mentioned)

A mention of the asthma study is present but it should also be added to the full text and in more detail.

The time range between the two methods of data collection is good, as 3 days (range 0-5 days) for the physical activity not to change. Were the participants visited at their homes for the face-to-face part? Could the data collection be described in more detail? Who called on the phone? Who asked the two questions face-to-face and how, where? Their professions, etc.

Lines 65 and 75; a clarification is needed, does the second contain IQR (interquartile range): "approximately 3 days (range 0-5 days)," and "median, 3 [1-4] days" in the text"

Line 84: Suddenly we switch to a tablet version: What is it? Face-to-face? So tablet should also be added to the data collection details. But were there, for example some visual items shown during the tablet version of data collection?

Table 1, ICC, 3rd line: 0.51(0.35, etc) with no space in between while the other cells have space before the parentheses.

line 90 "64years" needs a space. Spaces before parentheses should be checked and applied in a standard manner throughout the text.

Was there a difference according to sex?

Discussion, 1st paragraph, especially lines 104-105: The higher number of questions in the other physical activity scales could also render more accurate results with different modes of data collection.

Did the discrepancy augment with the time lag between the two modes of data collection? This could easily be analysed with their data...

Line 128 is a good suggestion to compare with the conventional method of data collection, and maybe the authors can apply it in the future. It is important for the readers to understand both from the abstract and the text that this study compares telephone interviewing to face-to-face tablet methods.

Lines 133-136: Could it be added to the conclusion that it is more reliable in especially some age groups? And could need some corrections, with estimations used for example from this study?

6. PLOS authors have the option to publish the peer review history of their article (what does this mean?). If published, this will include your full peer review and any attached files.

Reviewer #1: **Yes: **Eva Ekvall Hansson

Reviewer #2: **Yes: **Cátia Paixão

Reviewer #3: No

---

## [Author Response · Author response to Decision Letter 0]

23 Nov 2024

November 16th 2024

Reviewer #1: 

General comments

Thank you for giving me the opportunity to review this manuscript. The scope of the study is relevant to the readers of Plos One and the research field has relevance to assessment of physical activity. The study has a robust design, and the manuscript is generally well written, although blank spaces are missing here and there.

Title -Adequate

Abstract - Adequate

Introduction and aim 

The rationale for dividing the sample into age-groups needs to be addressed in the introduction and included in the aim. 

R: The rational for analysing the performance of BPAAT across three age groups has now been added to the Introduction section. Please see page 3, lines 56-58: “Furthermore, existing research on physical activity questionnaires has primarily focused on young and middle-aged adults, leaving a gap in understanding the psychometric properties of these tools in older adults (13, 14).”.

Please provide a reference for the first sentence in the introduction.

R: A reference has now been added. Please see page 3, lines 40-41: “Regular physical activity is a well-established protective factor in the prevention and management of the leading noncommunicable diseases, such as cardiovascular diseases, respiratory diseases and cancer (1).”.

Materials and methods 

State why the age-groups were chosen.

R: The rational for analysing 3 age groups has now been clarified in the Introduction section. In the methods section, we defined each age group. Please see page 4, lines 77-80: “Convergent validity (Spearman correlation coefficient, rho), reliability (intraclass correlation coefficient [ICC], Bland-Altman analysis), and agreement (% agreement, Cohen �) were determined for the all sample and across defined age groups (young adults 18-44 years, middle aged adults 45-64 years, older adults ≥65 years) (13, 14)”.

State the diary number for the ethical approval.

R: We have submitted all documentation related to ethical committee approval to PLOS One. Given that this was a multicenter study requiring approval from multiple regions, it was not feasible to include all references in this brief research communication, neither we felt comfortable in selecting one. We appreciate your understanding.

Results

Why is BMI for the group shown? Make sure only results that is included in the aim is shown, for example the group comparison.

R: BMI information has now been removed as suggested. Please see page 4, lines 86-87.

Page 4, line 78: state that figure 1 is figure 1a-d.

R: This has been modified accordingly.

Discussion - Well written

Conclusion – Adequate

Tables and figures – Adequate

References - Up-to date and relevant.

Reviewer #2: 

First of all, I would like to congratulate the authors for this interesting study. It is well-written, clear and addresses an important topic in the research field. I have however some amendments to improve the manuscript:

• Introduction, lines 38-39: “Regular… diseases.” Please add a reference that supports this quote.

R: A reference has now been added. Please see page 3, lines 40-41: “Regular physical activity is a well-established protective factor in the prevention and management of the leading noncommunicable diseases, such as cardiovascular diseases, respiratory diseases and cancer (1).”.

• Introduction, lines 48-49: “several clinical populations”. Please provide some examples of these clinical populations. The same for lines 52-53 “in patients with various health conditions”.

R: Examples have now been added in both sentences. 

Please see page 3, lines 40-41: “Regular physical activity is a well-established protective factor in the prevention and management of the leading noncommunicable diseases, such as cardiovascular diseases, respiratory diseases and cancer (1).”.

Please see page 3, lines 49-52: “Telephone interviewing has been shown to be comparable to face-to-face administration of self-report questionnaires in general older population and specific clinical populations, such as patients with musculoskeletal disorders and chronic obstructive pulmonary disease (8-10).”.

• Methods: were the population composed by health individuals, individuals with any medical condition or both? Please clarify this.

R: Epi-asthma study included adult subjects from the general population attending the participating primary care centers. We have now made that clear in the Methods section. Please see page 4, lines 64-66: “. Subjects from the general population were included if they were ≥18 years of age and were registered in the database of the participating primary healthcare centres located throughout mainland Portugal.”.

• Methods: Please clarify to the readers that the face-to-face version of BPAAT was applied via tablet.

R: We have now added this information in the Methods section. Please see page 4, lines 73-74: “During the face-to-face visit, participants completed the BPAAT on a tablet, with support available from a trained clinical researcher upon request.”.

• Methods: Regarding the individuals who performed the BPAAT via phone and in-person on the same day, how did the authors manage it? How long before the face-to-face visit was the phone call made? Could you clarify it? May be consider adding a figure…

R: The information provided in the Methods section has been improved. The two versions could be answered in the same day, with at least half day apart. Please see page 4, lines 69-71: “Subjects completed the Portuguese version of BPAAT (4) during a telephone interview and, after approximately 3 days (minimum half day, maximum 5 days), during a face-to-face visit in his/her primary care center.”.

• Methods: lines 69-71 – please include the interpretation used for the scores of rho, ICC and Cohen K.

R: Cut-offs for these psychometric properties have now been added in the Methods section. Please see page 4, lines 81-84: “Spearman correlation coefficient ranges from −1 to 1, with values approaching 0 showing the least correlation (16). The ICC ranges from 0 to 1 and ‘good’ to ‘excellent’ reliability can be considered for ICC>0.75 (17). The agreement level was interpreted as follows: slight (0.0-0.20), fair (0.21-0.40), moderate (0.41-0.60), substantial (0.61-0.80) or almost perfect (≥0.81) (17).”.

• Results: in figure 1, the authors divided the analysis by age decades. Could you explain the reason and clarify this division in the methods?

R: Age groups are now defined in the Methods section to clarify the reader. Please see page 4, lines 77-80: “Convergent validity (Spearman correlation coefficient, rho), reliability (intraclass correlation coefficient [ICC], Bland-Altman analysis), and agreement (% agreement, Cohen �) were determined for the all sample and across defined age groups (young adults 18-44 years, middle aged adults 45-64 years, older adults ≥65 years) (13, 14).”.

• Results: figure 1 and table 1 – In the paper from Cruz et al (2021), the results varied from gender. Even though the measures from Cruz et al and this paper are different (accelerometry vs BPAAT in Cruz and only BPAAT administered by different ways in this paper), did the authors consider analyzing the data by gender?

R: Thank you for your suggestion. Gender analyses were not included in the study for two main reasons. First, this was not one of our study objectives. Second, we believed that subdividing each age group by gender would result in insufficient statistical power to address the question adequately. We appreciate your understanding of our perspective.

Reviewer #3:

It is an interesting study and could be useful practically for researchers or professionals using this scale, and some other scales. It is also interesting to see the differences according to age. My specific comments are below:

Some of the references are written with a space after the text (like in line.46: "measures (Cleland") but some without space (like in l.47: "remotely(Pleguezuelos") thus they should be standardized, conforming the journal's requirements.

R: Citations have now been revised and standardized throughout the manuscript.

Line 47: "Telephone screening" could be described more openly, since it can also mean filling out a form by clicking its link/QR code etc on a smartphone...

R: Thank you for your suggestion. Explanation of telephone interview has now been added. Please see page 4, lines 69-73: “Subjects completed the Portuguese version of BPAAT (4) during a telephone interview and, after approximately 3 days (minimum half day, maximum 5 days), during a face-to-face visit in his/her primary care centre. Interviews were performed by a team of experienced interviewers (psychologists, nurses) using a Computer Assisted Telephone Interviews system (CATI).”.

Line 56: Aim: Which general population, it could be specified like all of Portugal or some parts of it, etc.

R: This research aim is relevant regardless of the context, which is why the word 'Portuguese’ was not included in the aim statement. However, this information is provided and clarified in the Methods section. Please see page 4, lines 64-66: “Subjects from the general population were included if they were ≥18 years of age and were registered in the database of the participating primary healthcare centres located throughout mainland Portugal.”.

Methods:

So the questionnaire was applied face-to-face, 3 days after the telephone interview. So the data were collected in this manner between May 2021-July 2023? Or were they contacted later on, like after November 2023, as the original study was on asthma? 

R: All data collection occurred between May and July 2023.

Was the ethical approval for the asthma study? Or the physical activity study? Could the ethical committee's decision numbers be provided? (I saw later detailed info in the attachment, ı wonder if just the names and number be stated somewhere but as it is on asthma, maybe the name of the study, the aim and localities with the no.s in parenthesescould be mentioned)

R: We have submitted all documentation related to ethical committee approval to PLOS One. Given that this was a multicenter study requiring approval from multiple regions, it was not feasible to include all references in this brief research communication, neither we felt comfortable in selecting one. We appreciate your understanding.

A mention of the asthma study is present but it should also be added to the full text and in more detail. 

R: Information about Epi-asthma has been added to the Methods section. Reader can also find the reference of the protocol study if interested in more information. Please see page 3, line 63: “EPI-ASTHMA was investigating the prevalence of current asthma in the adult population (15).”.

The time range between the two methods of data collection is good, as 3 days (range 0-5 days) for the physical activity not to change. Were the participants visited at their homes for the face-to-face part? Could the data collection be described in more detail? Who called on the phone? Who asked the two questions face-to-face and how, where? Their professions, etc.

R: The data collection range and local of data collection are now better described in the Methods section. Please see page 4, lines 69-74: “Subjects completed the Portuguese version of BPAAT (4) during a telephone interview and, after approximately 3 days (minimum half day, maximum 5 days), during a face-to-face visit in his/her primary care centre. Interviews were performed by a team of experienced interviewers (psychologists, nurses) using a Computer Assisted Telephone Interviews system (CATI). During the face-to-face visit, participants completed the BPAAT on a tablet, with support available from a trained clinical researcher upon request.”.

Lines 65 and 75; a clarification is needed, does the second contain IQR (interquartile range): "approximately 3 days(range 0-5 days)," and "median, 3 [1-4] days" in the text"

R: Indeed, IQR are inside square brackets. IQR has been added. Please see page 4, lines 86-87: “A total of 355 subjects (60.8% female) with a median of 54 [interquartile range IQR 42-66] years were analysed.”.

Line 84: Suddenly we switch to a tablet version: What is it? Face-to-face? So tablet should also be added to the data collection details. But were there, for example some visual items shown during the tablet version of data collection?

R: The data collection procedures are now better described in the Methods section, as answered above.

Table 1, ICC, 3rd line: 0.51(0.35, etc) with no space in between while the other cells have space before the parentheses.

R: This has been corrected. Please see table 1.

line 90 "64years" needs a space. Spaces before parentheses should be checked and applied in a standard manner throughout the text.

R: This has been corrected throughout the manuscript.

Was there a difference according to sex?

R: Gender analyses were not included in the study for two main reasons. First, this was not one of our study objectives. Second, we believed that subdividing each age group by gender would result in insufficient statistical power to address the question adequately. We appreciate your understanding of our perspective.

Discussion, 1st paragraph, especially lines 104-105: The higher number of questions in the other physical activity scales could also render more accurate results with different modes of data collection. 

R: We agree that the number of items could also have a role in explaining difference. This argument has been added. Please see page 6, lines 117-118: “The higher number of questionnaire items, consistent time intervals, and/or randomisation between the modes of administration may be potential reasons for these differences.”.

Did the discrepancy augment with the time lag between the two modes of data collection? This could easily be analysed with their data...

R: An exploratory analysis was performed (not shown) and we did not find a relationship between time lag and the reliability between the two modes. We believe this is related with the short time interval (maximum 5 days), with no significant time to change physical activity habits.

Line 128 is a good suggestion to compare with the conventional method of data collection, and maybe the authors can apply it in the future. It is important for the readers to understand both from the abstract and the text that this study compares telephone interviewing to face-to-face tablet methods.

R: Thank you for your comment. Data collection methods were clarified both in the Abstract and Methods section.

Lines 133-136: Could it be added to the conclusion that it is more reliable in especially some age groups? And could need some corrections, with estimations used for example from this study?

R: The caution in the oldest age group has been added in the study conclusions. Please see page 7, lines 145-147: “In conclusion, telephone administration of the BPAAT is a valid and reliable approach for monitoring of physical activity in the general population. However, it may slightly overestimate activity levels compared to face-to-face administration, particularly among subjects aged 65 years and older”.

---

## [Decision Letter · Decision Letter 1]

18 Dec 2024

PONE-D-24-41112R1Validation of the Brief Physical Activity Assessment Tool: Comparison of Telephone and In-Person AdministrationPLOS ONE

Dear Dr. Jácome,

Thank you for submitting your manuscript to PLOS ONE. After careful consideration, we feel that it has merit but does not fully meet PLOS ONE’s publication criteria as it currently stands. Therefore, we invite you to submit a revised version of the manuscript that addresses the points raised during the review process.

Please ensure that the final comments of the referees are addressed.

We look forward to receiving your revised manuscript.

Kind regards,

Filipe Prazeres, MD, MSc, Ph.D.

Academic Editor

PLOS ONE

Journal Requirements:

Reviewers' comments:

Reviewer's Responses to Questions

**Comments to the Author**

1. If the authors have adequately addressed your comments raised in a previous round of review and you feel that this manuscript is now acceptable for publication, you may indicate that here to bypass the “Comments to the Author” section, enter your conflict of interest statement in the “Confidential to Editor” section, and submit your "Accept" recommendation.

Reviewer #1: All comments have been addressed

Reviewer #2: All comments have been addressed

Reviewer #3: All comments have been addressed

2. Is the manuscript technically sound, and do the data support the conclusions?

Reviewer #1: Yes

Reviewer #2: Yes

Reviewer #3: Yes

3. Has the statistical analysis been performed appropriately and rigorously? 

Reviewer #1: Yes

Reviewer #2: Yes

Reviewer #3: Yes

4. Have the authors made all data underlying the findings in their manuscript fully available?

Reviewer #1: Yes

Reviewer #2: Yes

Reviewer #3: (No Response)

5. Is the manuscript presented in an intelligible fashion and written in standard English?

Reviewer #1: Yes

Reviewer #2: Yes

Reviewer #3: Yes

6. Review Comments to the Author

Reviewer #1: Thank you for your revised manuscript, which is much improved and now suitable for publication. Congratulations

Reviewer #2: The authors have addressed all my questions, thank you! I have no further comments. I would like to congratulate the authors for their brilliant job.

Reviewer #3: I would first like to thank the authors for the revision of the manusript and the point-by point explanations.

One analysis that could enrich the manuscript would be to analyze according to sex only; so separately for both sexes, one row for females and one row for males. Their activity levels might be quite different and their expressions also might be different. Seeing the results of these would be interesting.

7. PLOS authors have the option to publish the peer review history of their article (what does this mean?). If published, this will include your full peer review and any attached files.

Reviewer #1: No

Reviewer #2: No

Reviewer #3: **Yes: **Raika Durusoy

---

## [Author Response · Author response to Decision Letter 1]

20 Dec 2024

November 19th 2024

Comments to the Author

4. Have the authors made all data underlying the findings in their manuscript fully available?

Reviewer #1: Yes

Reviewer #2: Yes

Reviewer #3: (No Response)

Response: The data availability statement has been added during revision 1. All data supporting the findings are available from the Faculty of Medicine of University of Porto for all interested researchers who meet the criteria for access to confidential data. Because these data include patients’ personal information, the Ethical Committees that approved the study does not recommend that such data be made public unnecessarily. Please contact Cristina Jácome, the corresponding author at cjacome@med.up.pt, or the Faculty of Medicine of University of Porto at fmup@med.up.pt to request the data.

6. Review Comments to the Author

Reviewer #3: I would first like to thank the authors for the revision of the manuscript and the point-by point explanations. One analysis that could enrich the manuscript would be to analyze according to sex only; so separately for both sexes, one row for females and one row for males. Their activity levels might be quite different and their expressions also might be different. Seeing the results of these would be interesting.

Response: Gender analyses were not included in the study for two primary reasons. First, addressing gender differences was not one of our main study objectives. Second, we believed that subdividing each age group by gender would yield insufficient statistical power to adequately address the questions posed, and analyzing gender differences without considering age range did not align with our hypothesis that age would influence questionnaire responses. We appreciate your understanding of our perspective.

Nevertheless, we acknowledge the significance of investigating gender differences in physical activity, and we are currently preparing a manuscript that analyzes these data based on the questionnaire.

---

## [Editor Report · Decision Letter 2]

2 Jan 2025

Validation of the Brief Physical Activity Assessment Tool: Comparison of Telephone and In-Person Administration

PONE-D-24-41112R2

Dear Dr. Jácome,

We’re pleased to inform you that your manuscript has been judged scientifically suitable for publication and will be formally accepted for publication once it meets all outstanding technical requirements.

Kind regards, Feliz Ano Novo,

Filipe Prazeres, MD, MSc, Ph.D.

Academic Editor

PLOS ONE
---

## [Editor Report · Acceptance letter]

10 Jan 2025

PONE-D-24-41112R2 

PLOS ONE

Dear Dr. Jácome, 

I'm pleased to inform you that your manuscript has been deemed suitable for publication in PLOS ONE. Congratulations! Your manuscript is now being handed over to our production team.

Kind regards, 

on behalf of

Prof. Filipe Prazeres 

Academic Editor

PLOS ONE